# Role and Regulation of the RECQL4 Family during Genomic Integrity Maintenance

**DOI:** 10.3390/genes12121919

**Published:** 2021-11-29

**Authors:** Thong T. Luong, Kara A. Bernstein

**Affiliations:** Department of Pharmacology and Chemical Biology, UPMC Hillman Cancer Center, University of Pittsburgh School of Medicine, 5117 Centre Avenue, Pittsburgh, PA 15213, USA; thl27@pitt.edu

**Keywords:** RECQL4, genome stability, replication, telomere, double-strand break repair, base excision repair, mitochondria, nucleotide excision repair, DNA crosslink repair

## Abstract

RECQL4 is a member of the evolutionarily conserved RecQ family of 3’ to 5’ DNA helicases. RECQL4 is critical for maintaining genomic stability through its functions in DNA repair, recombination, and replication. Unlike many DNA repair proteins, RECQL4 has unique functions in many of the central DNA repair pathways such as replication, telomere, double-strand break repair, base excision repair, mitochondrial maintenance, nucleotide excision repair, and crosslink repair. Consistent with these diverse roles, mutations in *RECQL4* are associated with three distinct genetic diseases, which are characterized by developmental defects and/or cancer predisposition. In this review, we provide an overview of the roles and regulation of RECQL4 during maintenance of genome homeostasis.

## 1. RECQL4 Is Critical in Maintaining DNA Integrity and Disease Prevention

### 1.1. RECQL4 Is Unique Amongst the RecQ Helicases

Maintaining genomic stability is crucial for preserving genetic information and preventing disease. The stability of our genome is constantly threatened by exogenous and endogenous sources, such as ultraviolet radiation and reactive aldehydes, respectively. Our cells have developed an intricate system of proteins and pathways to counteract these insults. One such family of proteins is the evolutionary conserved RecQ helicases, consisting of RECQL1, RECQL4, RECQL5, Bloom syndrome protein (BLM), and Werner syndrome ATP-dependent helicase (WRN) (Figure 1A). This family of 3’ to 5’ DNA helicases are referred to as the “Guardians of the Genome” through crucial roles in DNA recombination, replication, and repair [1,2,3,4,5,6].

One critical member of this helicase family is encoded by the *RECQL4* gene, which is located on chromosome 8q24.3. The resulting 1208 amino acid protein has a conserved core helicase domain, however RECQL4 is quite different from the other RecQ helicases as it does not contain the conserved helicase and RNAse D C-terminal (HRDC) domain, which is needed for putative DNA binding (Figure 1A). Furthermore, RECQL4 has a strong DNA annealing activity, so much so that it was thought RECQL4 did not have helicase activity. However, RECQL4 helicase activity was later shown, when *in vitro* helicase assays were performed in the presence of excess oligonucleotide to prevent reannealing [8]. The RecQ helicases, RECQL1, BLM, and WRN are nuclear and RECQL5 is both nuclear and cytoplasmic. RECQL4 is unique in that it can be localized to the cytoplasm, nucleus, and mitochondria [6]. RECQL4 cytoplasmic localization is regulated by acetylation at K376, K380, K382, K385, K386 by p300 [9]. Moreover, RECQL4 is the only RecQ helicase with a mitochondrial localization signal (MLS) and is critical for upkeeping the mitochondrial genome (discussed below; Figure 1B).

Underscoring its many cellular functions, mutations in *RECQL4* are linked to three hereditary diseases: Rothmund-Thomson syndrome (RTS), RAPADILINO, and Baller-Gerold syndrome (BGS) (See Table 1 for all abbreviations used) [10,11]. Although these RECQL4-associated diseases share overlapping symptoms, there are distinct differences between them as described in the following sections and diagrammed in Figure 2. In addition, we discuss the unique cellular functions of RECQL4 and how alterations of specific RECQL4 functions contribute to human disease.

### 1.2. Rothmund-Thomson Syndrome

Rothmund-Thomson syndrome (RTS) is an autosomal recessive disorder that is extremely rare, with approximately 500 reported cases [12]. RTS is characterized by poikiloderma, developmental abnormalities, cataracts, premature aging, and increased cancer predisposition, particularly osteosarcomas. Symptoms of RTS were first described in 1868 by Rothmund and then later in 1936 by Thomson [13,14]. In 1957, Taylor decided to group these two findings and classified the disease as RTS [15]. One of the main features of RTS is chromosomal aberration including chromosomal rearrangements and aneuploidy [16,17,18]. Despite evidence for a genetic cause for RTS, the gene responsible for RTS remained elusive until the late 1990s. In 1998, the Shimamoto group cloned *RECQL4* in a study to find new human RecQ helicases [19]. A year later in 1999 the Furuichi group identified that a subset of RTS patients had mutations in *RECQL4* [20]. *RECQL4* mutations range from frameshift, splicing and nonsense, which commonly leads to a truncated helicase domain [10,11]. Two-thirds of RTS patients are associated with mutations in *RECQL4* and are referred to as RTS type II. The other third of RTS patients, referred to as RTS type I, are associated with mutations in another gene, anaphase promoting complex 1 (*ANAPC1*) or other yet to be identified gene(s) [21,22].

### 1.3. Baller-Gerold Syndrome

Similar to RTS, Baller-Gerold syndrome (BGS) is a rare autosomal recessive disease. It was originally described by Baller in 1950 and later by Gerold in 1959 [23,24]. The prevalence is currently unknown and fewer than forty cases have been reported [25,26,27,28]. The most common RECQL4 mutation observed in BGS patients is a C-terminal missense mutation, R1021W, and a frame shift mutation in exon 9 which deletes nucleotide number 2886 (2886 delta T frameshift mutation). BGS symptoms share similarities to RTS including poikiloderma, developmental and radial ray defects. However, unlike RTS patients, BGS is associated with coronal craniosynostosis. Alongside *RECQL4*, mutations in fibroblast growth factor receptor 2, *FGFR2*, and the transcription factor, *TWIST*, have also been implicated in the development of BGS [29,30,31].

### 1.4. RAPADILINO

RAPADILINO was first described in 1989 by Kaariainen et al., where they coined the name RAPADILINO after an acronym related to the patients’ syndromes. RA stands for radial ray malformations, PA for absent/hypoplastic patellae and cleft/high arched palate, DI for diarrhea and dislocated joints, LI for limb abnormalities and little size, and NO for slender nose and normal intelligence [32]. RAPADILINO is an autosomal recessive disease which shares similarities with RTS as it is characterized by short stature, radial ray defects, predisposition to osteosarcoma and lymphoma. However, unlike RTS patients, RAPADILINO is also associated with palatal abnormalities as well as joint dislocation. Unlike RTS and BGS, *RECQL4* is the only gene affiliated with RAPADILINO. Whereas RTS patients often have a truncated RECQL4 helicase domain (exon 8–14), the most common mutation in RAPADILINO leads to deletion of exon 7, which leaves the helicase domain intact but disrupts the second NLS of RECQL4 (Figure 1) [33]. Although the overall prevalence of RAPADILINO is unknown, it is most frequently observed in the Finnish population [31].

### 1.5. RECQL4 in Cancer

Loss of *RECQL4* function is associated with chromosomal instability, which is a driver of cancer [34,35]. A recent case study of five families determined that pathogenic mutation in the helicase domain of RECQL4 is highly associated with osteosarcoma. In contrast, RTS patients with mutations outside that helicase domain develop mild symptoms and do not develop cancer [36]. This suggests that the helicase function of RECQL4 is critical for maintenance of genomic stability and subsequent cancer prevention. However, there was a rare case where a RAPADILINO patient had a mutation outside the helicase domain but still developed lymphoma [33]. Although patients with *RECQL4* loss of function are predisposed to osteosarcomas, *RECQL4* overexpression is also observed in several cancers including breast, cervical, gastric, oral, osteosarcoma, and prostate [37,38,39,40,41,42,43]. *RECQL4* is associated with mediating different survival factors to promote cancer growth such as the apoptosis inhibitor, SURVIVIN, and the tumor suppressor, p53. SURVIVIN interacts with RECQL4 and *RECQL4* knockdown results in reduced SURVIVIN expression [39]. RECQL4 also interacts with p53 and can sequester its transcriptional activity, ([44]; see the mitochondria section, Table 2). Consistent with promotion of tumor cell growth, *RECQL4* overexpression is correlated with tumor aggressiveness in breast and prostate cancer and its suppression results in reduced proliferation [38,42]. Strikingly, in a prostate cancer focused study, suppressing *RECQL4* to levels observed in non-malignant human epithelial cells is sufficient to reduce invasive growth and tumorigenic potential [38]. Another strategy that cancer cells use to survive is through the upregulation of drug efflux pumps, which reduce the intracellular concentration of toxic chemotherapeutics [45]. Strikingly, RECQL4 transcriptionally activates the efflux pump, MDR1, driving its expression to mediate cisplatin resistance [43]. Besides its role in modulating proteins critical for cellular survival and drug resistance, RECQL4 can also support cancer progression and survival by its replicative and DNA repair roles as discussed below. Intriguingly, *RECQL4* is located on chromosome 8q24, which is a particular hotspot for overexpression in cancer as it is near the oncogene, c-Myc [46,47]. It is unclear whether *RECQL4* upregulation is simply a byproduct of being near c-Myc or purely coincidental. However, it is rare for one gene to play so many advantageous roles for cancer cell proliferation and survival.

## 2. Model Systems to Study RECQL4 Function

### 2.1. Due to Difficulties Studying RECQL4 Function in Mice, Other Model Systems Have Been Used

Unlike the other two diseases associated with RecQ helicase dysfunction, BLM and WRN, RECQL4 has not been as well characterized. This is because RECQL4 contains a Sld2-like domain, which is crucial for replication initiation, in the N-terminus (discussed further below). Consistent with this, knockout of *RECQL4* leads to embryonic lethality in *D. melanogaster* and *M. musculus* [59,60]. To circumvent the embryonic lethality observed in a complete knockout system, additional mouse models were created to delete specific regions of RECQL4. A mouse model lacking exon 13, which encodes the latter portion of the helicase domain, resulted in five percent viable pups. However, those surviving animals do not have poikiloderma and therefore do not fully recapitulate RTS [61]. An additional mouse model which lacks most of the helicase domain (exons 9–13) resulted in more than eighty percent viable pups and more closely resembles RTS patients, including poikiloderma [62]. Due to the prior difficulty in making viable mouse models, other model systems have also been used to study the function of RECQL4 including frogs (*Xenopus laevis*), flies (*Drosophila melanogaster*), and yeast (*Saccharomyces cerevisiae*) and are described below.

### 2.2. Yeast Has Been Critical in Studying the Functions of RecQ Helicases

RecQ helicases were first discovered in *E. coli* and later a yeast homolog, *SGS1*, was uncovered [63,64]. For 14 years, Sgs1 was thought to be the only RecQ helicase in yeast as it has conserved functions with other human RECQ protein family members, such as WRN and BLM. However in 2008, Hrq1 was discovered through a computational screen by Barea et al. [65]. In this study, hydrophobic cluster analysis was used to find yeast genes that have similar hydrophobic regions to RECQL4, which could indicate similar quaternary structure. The revealed Hrq1 as a potential a RECQL4 homolog. Since then, Hrq1 has been validated as a bonafide homolog of RECQL4 due to its conserved function in genome integrity. For example, *in vitro* biochemical studies demonstrate that Hrq1 and RECQL4 bind and unwind similar DNA substrates including bubbles, D-loops, and poly(dT) forks [66]. The primary difference between yeast Hrq1 and human RECQL4 is that yeast encodes a separate Sld2 protein whereas human RECQL4 is a fusion between Sld2-like domains at the N-terminus of the helicase (Figure 1). Below we discuss the role and regulations of RECQL4 during genomic maintenance as well as highlight the crucial role that model systems have played in teasing out the functions of RECQL4 (Figure 3 summarizes the diverse functions of RECQL4 discussed).

## 3. Conserved Role for the RECQL4 Family during DNA Replication

### 3.1. Xenopus RECQL4 Role in Replication Initiation

*Xenopus* egg extracts served as a powerful *in vitro* model system to study genome maintenance. Since cellular extracts contain all the factors necessary to perform genome maintenance, it allows researchers to systematically immunodeplete specific proteins and study their function [67,68,69]. *Xenopus* has been critical in initial investigations of RECQL4 function. For example, RECQL4 function during replication was first described in *Xenopus laevis* by the Venkitaraman group. The first clues that RECQL4 had a replication function was shown by Sangrithi et al. who observed that both the N-terminus of human and frog RECQL4 have sequence similarity to the essential yeast replication initiation factor, Sld2 [70]. They demonstrated that immuno-depletion of *Xenopus* RECQL4 (called xRTS) impairs replication initiation and nascent DNA synthesis in the *Xenopus* egg extract system. xRTS is loaded on chromatin following recruitment of the pre-replication complex and is needed for loading of the ssDNA binding protein, RPA, as well as the assembly of DNA polymerase α (Figure 3A). Interestingly, the authors show that expression of only the N-terminal Sld2-like domain only led to 20% rescue whereas the full-length protein is needed for complementation of depleted xRTS. Importantly, this replication defect in *Xenopus* extracts is complemented with recombinant human RECQL4. RECQL4 replication function is dependent on its helicase activity as mutating the conserved Walker B motif (RECQL4-D605A), needed for ATP hydrolysis, does not rescue a xRTS immuno-depleted extract. These results suggest that there are multiple regions of RECQL4 needed for replication initiation. Similar findings were also observed by Matsuno et al. who demonstrated that N-terminal domain of xRTS physically interacts with *Xenopus* Cut5 (TOPBP1 in humans), an essential replication initiation factor (Table 2) [48]. The interaction between Cut5 and xRTS is independent of the phosphorylation status of xRTS on its N-terminal Sld2-like domain. Unlike *Xenopus*, the human ortholog of Cut5, TOPBP1, does not directly interact with RECQL4 [49]. Therefore, there are likely key differences between *Xenopus* and human RECQL4 family members.

### 3.2. Drosophila RECQL4 Role in Replication Initiation

Like *Xenopus*, *D. melanogaster* RECQL4 is also critical for DNA replication and *RECQL4* knockout results in embryonic lethality and failure in cellular proliferation [60]. Although knockout of *RECQL4* leads to embryonic lethality in flies, due to the powerful genetic tool of mosaicism researchers can study how *RECQL4* knockout effects individual cells [60,71]. Importantly, similar to the *Xenopus* study, *Drosophila* RECQL4 helicase function is critical for replication as helicase dead *Drosophila* RECQL4 is not able to rescue the proliferation defect seen in null RECQL4 [72]. These results suggest that the RECQL4 family has a conserved replication function in replication initiation.

### 3.3. Human RECQL4 Role in Replication Initiation

Like frogs and flies, subsequent studies demonstrate that human RECQL4 has a critical role during replication initiation through loading the replicative helicase, CMG complex (CDC45-MCM2-7-GINS) (Figure 3A; [49,73]). Using bimolecular fluorescence complementation assays as well as chromatin fractionation, Im et al. showed that knockdown of *RECQL4* leads to decreased interaction between CMG complex members resulting in its failure to load on chromatin [73]. RECQL4 was subsequently found by Xu et al. to interact with CMG complex members, MCM7 and SLD5, and another important replication initiation protein, MCM10 (Figure 1B, Table 2) [49]. RECQL4 interaction with MCM7 and SLD5 is MCM10 dependent. Consistent with a role during replication initiation, RECQL4 interaction with the MCM7, SLD5, and MCM10 is enriched during G1/S. Furthermore, si*RECQL4* knockdown leads to decreased DNA synthesis [42,49]. Lastly, they demonstrated that MCM10 plays a critical role in regulating RECQL4 function by inhibiting its helicase activity *in vitro*. MCM10 and subsequent inhibition of RECQL4, depends on RECQL4 phosphorylation state, as phospho-mimetic RECQL4 mutant (S89E, T93E, T139E) fails to interact MCM10 [49]. The authors postulated that cyclin dependent kinase, CDK, mediated phosphorylation may regulate RECQL4 function in a cell cycle dependent manner. Later studies demonstrated that CDK-mediated phosphorylation can indeed modulate RECQL4 activity, albeit during DSB repair [74]. It remains to be determined whether CDK-mediates RECQL4 replicative functions.

## 4. Role of Hrq1 and RECQL4 during Telomeric DNA Maintenance

### 4.1. Human RECQL4 Role in Telomeric DNA Maintenance

There are many clues that suggest that RECQL4 has an important role in telomere maintenance. First, RTS is a progeroid syndrome, which is often associated with telomere dysfunction. Similarly, RTS patients exhibit symptoms analogous to dyskeratosis congenita, another disease caused by telomere dysfunction [61,75]. Similar to RTS patients, mutations in another RecQ helicase, WRN, results in Werner syndrome. Werner patients share clinical features with RTS patients such as developmental abnormalities and premature aging [76,77,78]. It is thought that Werner syndrome symptoms are largely due to WRN’s crucial function during telomere maintenance [5,79]. Therefore, it was hypothesized that RECQL4 has an important role during telomere maintenance.

In 2012, Ghosh et al. demonstrated that RTS cells have more fragile telomeres compared to age/sex matched normal fibroblasts by telomeric fluorescent in situ hybridization (FISH) [50]. Similar to RTS cells, *RECQL4* knockdown also increases telomere fragility, telomere sister chromatid exchanges, and telomere dysfunction-induced foci (TIFs). TIFs are indicative of a DNA damage response occurring at the telomere [80]. TIFs are characterized by 53BP1 colocalization with the telomere binding and Shelterin complex component, TRF1. These findings demonstrate that RECQL4 protects telomeres from DNA damage.

RECQL4 localizes to telomeres, where it interacts with multiple members of the Shelterin complex (Figure 3B, Table 2). The strongest interaction is between RECQL4 and TRF2. RECQL4 interaction with TRF1 and TRF2 stimulates RECQL4-dependent D-loops resolution at both undamaged and oxidized telomeres *in vitro*. Since both RECQ helicases, RECQL4 and WRN, have critical telomere functions, it was hypothesized that WRN and RECQL4 cooperatively function. Indeed, RECQL4 physically and functionally interacts with WRN (Table 2). RECQL4 stimulates WRN-dependent D-loop unwinding at both undamaged and oxidized telomeric *in vitro*. Further validating their roles in protecting telomeres from damage, both RECQL4 and WRN resolve oxidized telomeric D-loops better than undamaged D-loops [50]. The Bohr group also later demonstrated that RECQL4 can also unwind telomeric D-loops which contain thymine glycol, a potent replication fork stalling lesion, with even a greater affinity than telomeric D-loop containing 8-oxoG. Interestly, in contrast with the telomeric D-loop containing 8-oxoG, they observed that RECQL4 does not stimulate WRN ability to resolve telomeric D-loops with thymine glycol [81]. Together, RECQL4 has crucial roles in the repair of damage at telomeres and telomeric DNA replication.

### 4.2. Yeast Hrq1 Role during Telomeric DNA Maintenance

Recent studies in yeast also demonstrate a conserved role for Hrq1 in telomere maintenance. Bochman et al. showed that Hrq1 binds telomeres via chromatin-immunoprecipitation [82]. Hrq1 functions to inhibit telomerase as *hrq1Δ* cells have de novo telomere addition at DSB ends. In vitro experiments also show that high concentrations of Hrq1 stimulate telomerase extension activity on longer telomere substrates [83]. The authors postulate that these contradictory findings are due to the synergistic functions of both the DNA helicase Pif1 and Hrq1 to inhibit telomerase. Indeed, addition of Pif1 to the reaction in equimolar concentration with Hrq1 leads to inhibition of telomere extension *in vitro*. It remains to be determined whether increased Hrq1 concentration relative to Pif1 is sufficient to drive telomerase activity. It is unclear whether human RECQL4 has any effect on telomerase. While telomerase is not generally active in untransformed cells, immortalized cancer cells upregulate telomerase. Since RECQL4 is often overexpressed in cancer cells, it would be critically important to determine whether RECQL4 has a stimulatory or inhibitory effect on telomerase in cancer.

## 5. Role of RECQL4 during DNA Double-Strand Break Repair

### 5.1. Human and Xenopus RECQL4 Role during Double-Strand Break Repair

DNA double-strand breaks (DSB) are considered the most lethal type of DNA damage, as single, unrepaired DSB results in cell death [84,85]. Early studies suggested that RECQL4 may function during DSB repair since a subset of RTS patients are sensitive to the DSB inducing agent, γ-irradiation [76,86]. RECQL4′s role in DSBR was further collaborated by the Stagljar group who demonstrated that upon treatment with the topoisomerase II inhibitor, etoposide, RECQL4 colocalizes with ssDNA and RAD51 foci [54]. Furthermore, RECQL4 interacts with RAD51 by co-immunoprecipitation (Table 2). These early studies demonstrated that RECQL4 functions with RAD51 to repair DSBs.

RECQL4 function in DSB repair is likely conserved as the Enomoto group demonstrated that its *Xenopus* homolog, xRTS, localizes to chromatin upon DSB induction [87]. Furthermore, xRTS loading depends on the DNA damage checkpoint proteins, ataxia-telangiectasia mutated (ATM) and DNA-dependent protein kinase, DNA-PK, but not RAD51. These findings suggest that xRTS may function upstream of RAD51 and perhaps to facilitate its loading. Consistent with xRTS role in DSB repair, the authors demonstrated that xRTS depletion led to prolonged γ-H2AX foci compared to mock-treated egg extracts. These early results from mammalian cells and *Xenopus* extracts were later corroborated by the Bohr’s group. In a 2010 study, Singh et al. found that RECQL4 was recruited laser-induced damage sites, where it colocalized with 53BP1 and γ-H2AX, by time-lapse microscopy [88]. Similar to *Xenopus*, *RECQL4* deficient human cells delay DSB repair as prolonged 53BP1 foci are observed. However, unlike the Xenopus, human RECQL4 is still recruited to damage sites, even in the absence of ATM. RECQL4 recruitment to DSB sites depends on its nuclear localization. Interestingly, it was shown that RECQL4 retention at DSB sites is diminished in the absence of another RecQ helicase, BLM, suggesting cooperativity between these two proteins (Figure 1B, Table 2) [51]. Although these studies suggested that RECQL4 functions during DSB repair, its precise role has remained enigmatic until recently.

### 5.2. RECQL4 Role during Non-Homologous End Joining

RECQL4 has roles during the two major DSB repair pathways, non-homologous end joining (NHEJ) and homologous recombination (HR) (Figure 3C; [52,53,74,89]. In 2014, Shamanna et al. demonstrated that RECQL4 knockdown leads to decreased end joining both in *in vitro* and *in vivo* [52]. Cellular extracts from *RECQL4* knockdown cells have decreased capacity to join DNA ends *in vitro*. This was recapitulated *in vivo* as *RECQL4* knockdown leads to decreased NHEJ using a GFP reporter system and similar results were later confirmed by others [52,74,89]. Shamanna et al. also demonstrated that RECQL4 physically and functionally interacts with the key NHEJ protein, KU70, in the N-terminus of RECQL4. Furthermore, addition of RECQL4 increases KU complex DNA binding *in vitro* [52]. KU complex interaction with RECQL4 was later confirmed *in vivo* (Table 2) [74,89]. Lu et al. demonstrated that RECQL4 interacts with KU70 in a cell cycle dependent manner with increased interaction during the G1 stage, when NHEJ is the predominant DSB repair pathway (Figure 3C) [74,90]. Tan et al. later demonstrated that *RECQL4* knockdown leads to decreased recruitment of KU80 to DSB sites [89]. These results suggest that RECQL4 works cooperatively with the KU complex to mediate NHEJ.

### 5.3. RECQL4 Role during Homologous Recombination

RECQL4 role during HR was first shown by Lu et al. where RECQL4 was found to promote DNA end resection, a critical step during early HR (Figure 3C) [53]. RECQL4 physically and genetically interacts with the key end resection proteins, MRE11 and CtIP. RECQL4 functions before CtIP but after MRE11. A later study demonstrated that the interaction between MRE11 and RECQL4 is mediated by RECQL4 phosphorylation (S89/S251) by CDK1 and CDK2 during S/G2 cell cycle phase [74]. Phosphorylation of RECQL4 stimulates its helicase activity *in vitro*. RECQL4 helicase activity is critical for its function as a catalytically inactive mutant results in impaired end resection and HR [53]. Together, these findings demonstrate that RECQL4 functions with MRE11 to promote DNA end resection during HR.

### 5.4. Ubiquitylation of RECQL4 during DSB Repair

Interestingly there are conflicting findings on the role of RECQL4 ubiquitylation during DSB repair. Lu et al. observed that RECQL4 is ubiquitylated by DDB1-CUL4A following CDK1/CDK2 phosphorylation at S89 and S251. Ubiquitylation of RECQL4 results in increased RECQL4 recruitment to chromatin and sites of laser-induced DSBs [74]. However, a later study by Tan et al. demonstrated that RECQL4 is ubiquitylated at K876, K1048, and K1101 by RNF8 with the help of WRAP53β. In this case, RECQL4 ubiquitylation led to dissociation of RECQL4 from DSBs sites [89]. The second study did not interrogate whether ubiquitylation of RECQL4 by RNF8 is facilitated by CDK1/CDK2 phosphorylation or not. Furthermore, the study by Lu et al. did not determine where DDB1-CUL4A ubiquitylates RECQL4. Thus, it is possible that the ubiquitylation sites are different, where ubiquitylation by DDB1-CUL4A leads to accumulation of RECQL4 whereas RNF8-mediated ubiquitylation leads to its dissociation. It is also possible that DDB1-CUL4A and RNF8 compete to dynamically regulate RECQL4 at DSB sites and additional studies are needed to differentiate between these possibilities.

## 6. Role of RECQL4 during Base Excision Repair

### 6.1. RTS Cells Are Sensitive to Oxidizing Agents

Base excision repair (BER) mediates repair of base damage that results from DNA methylation and oxidization as well as other types of base damage. The Hock group first hypothesized a potential role for RECQL4 during repair of base damage by postulating that the cataracts observed in RTS patients could be due to misrepair of oxidative damage in the eye lens. The authors showed that RECQL4 forms nuclear foci following exposure to the oxidative damaging agent, hydrogen peroxide (H_2_O_2_) [91]. Consistent with a function for RECQL4 in BER, RTS cells increase the amount of 8-oxo-guanine following H_2_O_2_ treatment compared to control cells. This increase in 8-oxo-guanine in RTS cells also correlates with decreased cell viability as well as more senescent cells.

### 6.2. RECQL4 Localizes to Sites of Oxidized DNA Damage

Further evidence supporting a function for RECQL4 during BER was found by the Frank group who analyzed RECQL4 localization following exposure to a variety of DNA damaging agents. Utilizing a plasmid encoding GFP tagged RECQL4 and live cell imaging, they found that RECQL4 is primarily nuclear and identified its nuclear localization sequence (NLS) (Figure 1B) [57]. Consistent with a function for RECQL4 in BER, RECQL4 relocalizes into distinct nuclear foci upon exposure to the oxidative DNA damaging agents, H_2_O_2_ and streptonigrin. Furthermore, RECQL4 foci colocalizes with key BER proteins such as apurinic (AP) endonuclease 1, APE1, and the flap endonuclease, FEN1 (Table 2) [55]. Suggesting that the co-localization between RECQL4 and BER proteins is functionally important, RECQL4 stimulates APE1 endonuclease as well as FEN1 incision activities *in vitro* (Figure 3D). Although no colocalization was observed between RECQL4 and the key BER polymerase, Pol β, RECQL4 can also stimulate DNA synthesis by Pol β *in vitro* (Figure 3D) [55].

### 6.3. RECQL4 Interacts with Key BER Proteins

Suggesting a direct interaction between RECQL4 and BER proteins, RECQL4 immunoprecipitated with the 8-oxoguanine-glycosylase, OGG1, as well as a core member of the BER pathway, PARP1, through its C-terminus (Figure 1B; [56,57]). Similar to APE1, FEN1, and Pol β, RECQL4 stimulates the AP lyase activity of OGG1 *in vitro* (Figure 3D). RECQL4 interaction and subsequent stimulation of OGG1, upon oxidative stress, is induced by RECQL4 acetylation. RECQL4 acetylation is mediated by CREB-binding protein, CBP, which acetylates not only the five acetylation sites mentioned above (K376, K380, K382, K385, K386) but other undetermined sites as well. Deacetylation of RECQL4 is mediated by deacetylase SIRT1. Suggesting that SIRT1 plays a key role in mediating RECQL4 activity during BER, RECQL4 deacetylation results in a decreased interaction between RECQL4 and OGG1 [56]. Therefore, RECQL4 acetylation plays a key role in regulating its function during BER. Loss of RECQL4 results in misregulation of BER genes, as RTS cells fail to upregulate BER genes in response to oxidative damage by microarray analysis [55]. Together, these findings demonstrate that RECQL4 has a critical role during BER by stimulating the enzymatic activities of critical BER proteins, and that in its absence cells cannot properly activate an oxidative damage response.

## 7. Role of RECQL4 during Nucleotide Excision Repair

Nucleotide excision repair (NER) mediates the repair of bulky DNA adducts such as 6-4 photoproducts and cyclobutane pyrimidine dimers (CPD), which result from UV exposure [92,93]. Patients with defects in the canonical NER genes suffer from an inherited disorder called Xeroderma pigmentosum (XP). XP symptoms include profound sensitivity to the sun and increased incidence of skin lesions and cancers [92,93]. Multiple studies have implicated RECQL4 to have an important role during NER. For example, early studies reported that a subset of RTS patients were photosensitive, alongside developing erythema and other skin abnormalities [76,94]. Later reports demonstrated that cells from RTS patients have increased UV sensitivity, which is consistent with a role for RECQL4 during NER [95,96,97]. In a 2008 study by the Luo group, RECQL4 was found to relocalize into nuclear foci following UV irradiation [58]. Importantly, they showed that complementing RTS fibroblast cells with RECQL4 led to faster resolution of CPDs and increased survival following UV compared to the uncomplemented RTS cells. To further validate RECQL4 role in CPD resolution, the Luo group observed that normal fibroblast cells resolved CPDs more slowly when RECQL4 is knocked down by siRNA. Suggesting a direct role for RECQL4 in NER, RECQL4 interacts and colocalizes with XPA, a critical NER protein, following UV treatment (Figure 3E, Table 2). XPA functions to verify the nucleotide damage and following which the DNA is unwound to facilitate DNA excision [92,93]. RECQL4 functions downstream of XPA, as RECQL4 recruitment into UV-induced foci is diminished in XPA-null cells. Despite these studies, the exact function of RECQL4 during NER remains ambiguous. Thus, it is possible that RECQL4 could act similarly to XPB, 3′ to 5′ DNA helicase, unwind the DNA following XPA damage verification. Alternatively, RECQL4 could function as a backup to XPB. Since 6-4 photoproducts and CPDs can block replication fork progress, perhaps RECQL4 could initiate replication to rescue forks stalled by UV-induced damage.

## 8. Role of Hrq1 and RECQL4 during DNA Crosslink Repair

DNA crosslinks are extremely deleterious lesions as they can stall transcription and replication, and can hinder recruitment of proteins involved in DNA maintenance [98,99,100]. Unrepaired DNA crosslinks result in mitotic catastrophe and, ultimately, cell death [101]. DNA crosslinking agents, such as mitomycin C and cisplatin, are used as chemotherapeutics. Besides these exogenous sources, there are also endogenous sources of DNA crosslinking agents such as reactive aldehydes [102,103,104,105]. Reactive aldehydes are formed through normal metabolism and are extremely reactive and can interact with DNA to form crosslinks. An early study examining the sensitivity of RTS cells to different genotoxic agents, found that RTS cells exhibit increased sensitivity to cisplatin [97]. Recent studies with mammalian cell lines demonstrated that *RECQL4* knockdown results in increased cisplatin sensitivity [42,43]. This suggested that RECQL4 may have a role during DNA crosslink repair. It was later confirmed that the RECQL4 family has a conserved role during DNA crosslink repair, as knockout of *HRQ1* in yeast results in increased cisplatin and MMC sensitivity [82,106]. Furthermore, Hrq1 helicase function is necessary for crosslink repair as catalytically inactive mutant Hrq1-K318A, is more sensitive to both cisplatin and MMC [82]. Later genetic studies by the Bochman group determined that Hrq1 functions downstream of the exonuclease, Pso2 (SNM1 in mammals), during crosslink repair but its exact function during crosslink repair has yet to be determined in either yeast or mammals [66].

## 9. Role of RECQL4 in Mitochondrial Maintenance

### 9.1. RECQL4 Localizes to the Mitochondria

Unlike WRN and BLM which are primarily nuclear, RECQL4 is localized to the cytoplasm as well as the nucleus. This is particularly intriguing because RECQL4 is the only RecQ helicase to have a mitochondrial localization sequence (Figure 1B, [44,107]). Furthermore, one of the symptoms implicated with *RECQL4* dysfunction is a progeroid syndrome. Alongside telomere shortening, another major contributor to aging is mitochondrial dysfunction [108,109,110,111,112]. Therefore, RECQL4 was hypothesized to have a role in mitochondrial maintenance. The Bohr group demonstrated that human RECQL4 localizes to the mitochondria both by immunofluorescence and cellular fractionation [113]. Furthermore, they showed RECQL4 depletion leads to increased mitochondrial DNA damage and decreased oxygen consumption rates. Together, these findings suggest that RECQL4 has a critical role in protecting the mitochondrial genome.

### 9.2. RECQL4 Mitochondrial Localization Is Critical for Sequestering p53 and Enhancing Mitochondrial DNA Replication

Later studies by the Sengupta and Zhao groups confirmed RECQL4 mitochondrial localization [44,107,114] and RECQL4 interaction with the mitochondria importer, TOM20, was later described by De et al. [44]. Intriguingly though, RECQL4 protein is undetectable in the mitochondria during S phase, likely due to its role in DNA replication [44]. RECQL4 mitochondrial localization is important not only for RECQL4 function but also to recruit the tumor suppressor gene, p53, into the mitochondria under unperturbed conditions (Figure 1B and Figure 3F, Table 2). RECQL4 and p53 colocalization to the mitochondria enables mitochondria polymerase, Polγ, to mediate mtDNA synthesis [44,115]. Another important function for the colocalization between RECQL4 and p53 in the mitochondria is to sequester a portion of p53 away from the nucleus, thus inhibiting its transcriptional activity [44]. However, upon DNA damage, the interaction between RECQL4 and p53 is lost and both proteins relocalize to the nucleus to mediate the DNA damage response (Figure 3F).

### 9.3. Cells Deficient in RECQL4 Mitochondrial Localization Has Perturbed Bioenergetics

Later studies corroborated the importance of RECQL4 for mitochondrial maintenance, as disruption of RECQL4 MLS results in bioenergetics dysfunction indicated by increased aerobic glycolysis [107]. The shift to utilizing aerobic glycolysis is called the Warburg effect and is commonly observed in cancer cells [116]. It was hypothesized that the increased aerobic glycolysis caused by loss of RECQL4 mitochondrial localization could drive tumorigenesis. Consistent with this notion, cells with defective RECQL4 mitochondria localization have increased invasive capability [107]. RECQL4 is critical for mitochondrial maintenance but its exact role remains elusive. Unlike other cellular compartments, mitochondria are more susceptible to oxidative DNA damage as oxygen radicals are constantly released during ATP production [117,118]. BER plays a key role in the repair of oxidative damage in mitochondrial DNA [119,120]. Therefore, one possibility is that RECQL4 may function to enhance the BER polymerase, Pol β activity, in the mitochondria as has been shown in the nucleus [55]. Future studies are needed to pinpoint the exact function of RECQL4 in mitochondrial genome maintenance.

## 10. Conclusions and Future Directions

In this review, we highlighted the multifaceted roles that the RECQL4 family have in maintaining genome stability. With a plethora of functions, it is clear how disruption of RECQL4 can lead to different devastating and diverse diseases. However, we are only beginning to understand the functions of RECQL4 in these different pathways. For example, although the RECQL4 family has a role in DNA crosslink repair, its role during this process remains ambiguous. The repair of DNA crosslinks is a very intricate process which requires proteins from a multitude of different pathways. Since DNA crosslinking agents are used as chemotherapeutics, understanding how RECQL4 repairs crosslinks could be informative for clinicians. While RECQL4 is critical for genome maintenance and subsequent disease/cancer prevention, overexpression of RECQL4 is observed in a variety of different cancers. It is still unclear whether RECQL4 itself drives tumorigenesis and this will be an exciting new area for future study.

## Figures and Tables

**Figure 1 genes-12-01919-f001:**
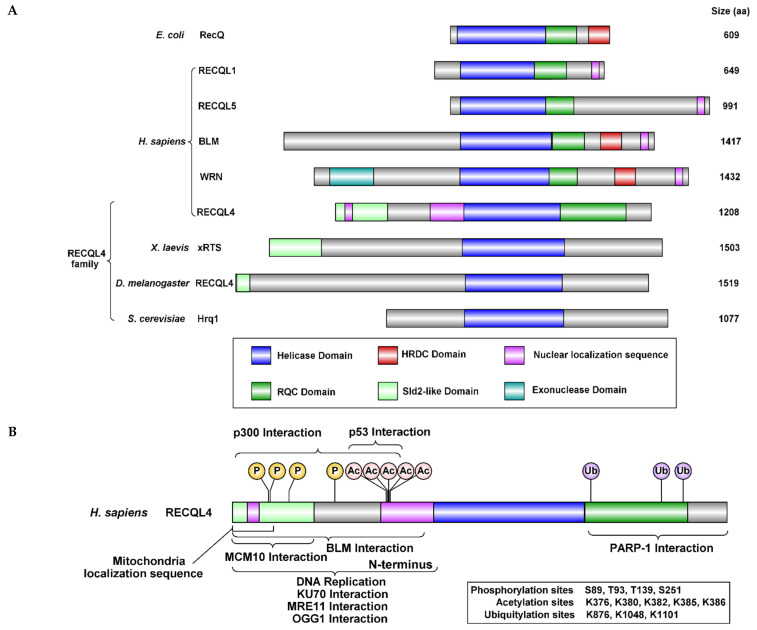
Structural features of RecQ helicases. (**A**) The RecQ proteins have domains that are conserved from bacteria through humans. The core helicase domain (dark blue box) is found throughout the protein family. However, unlike the other members of the RECQ family, RECQL4 is missing both the RQC (dark green box) and HRDC (red box) domains. With the exception of yeast, the RECQL4 family (including *H. sapiens*, *X. laevis*, *D. melanogaster*) have a conserved Sld-like domain, which is crucial replication initiation (light green box). In yeast, Sld2 is encoded as a separate protein. The nuclear localization sequence is indicated by a purple box and the exonuclease domain of WRN with a light blue box. (**B**) Interaction domains and post-translational modified regions of RECQL4. The mitochondrial localization sequence is located in the first 100 amino acid of RECQL4. Protein interactions with RECQL4 with BLM, MCM10, KU70, MRE11, OGG1, p300, p53, and PARP-1 are indicated by brackets. Phosphorylation sites (S89, T93, T139, S251) are indicated by a yellow circled P, acetylation sites (K376, K380, K382, K385, K386) by pink circled Ac, and ubiquitylation sites (K876, K1048, K1101) by purple circled Ub. Created by using Illustrator for Biological Sequences (IBS) [7].

**Figure 2 genes-12-01919-f002:**
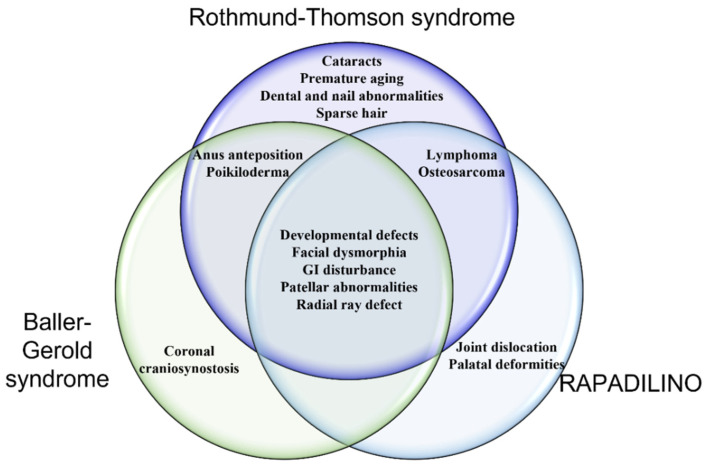
Venn diagram illustrating the similarities and differences between the three *RECQL4* related diseases.

**Figure 3 genes-12-01919-f003:**
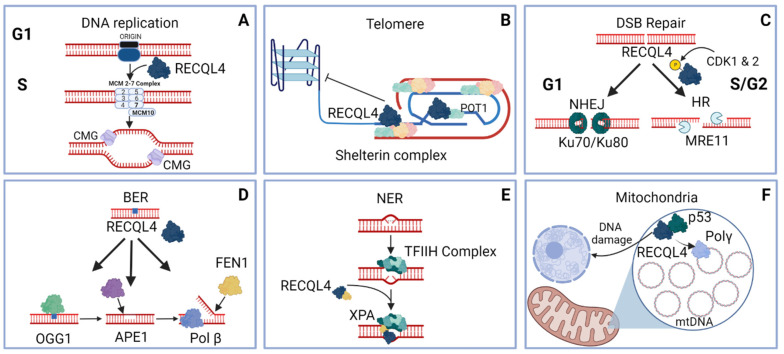
RECQL4 roles in genome maintenance. (**A**) RECQL4 (dark blue) is critical for replication initiation, RECQL4 is loaded following the pre-RC to mediate loading of DNA polymerase α and the CMG complex (light purple). The origin is indicated by a black and dark blue boxes and the MCM complex, consisting of MCM2-7, is shown with light blue boxes and MCM10 in a light blue box as well. (**B**). RECQL4 interacts with the Shelterin complex [POT1-TRF1-TRF2 (teal-beige-pink proteins, respectively)] to resolve D-loops (blue and red DNA structure on right) and G-quadruplexes (light blue structured DNA on left). (**C**). RECQL4 participates in DSB repair in a cell cycle dependent manner. RECQL4 functions with the KU complex (green circle) during G1 to mediate DNA end joining (NHEJ). During S/G2, RECQL4 cooperates with MRE11 (light blue pacman) to promote DNA end resection during homologous recombination (HR). (**D**). RECQL4 stimulates the function of multiple BER proteins at different stages. At the initial stage of damage recognition, RECQL4 stimulates OGG1 (light green protein) lyase activity, subsequently RECQL4 stimulates APE1 (purple protein) endonuclease activity, and finally RECQL4 stimulates FEN1 (yellow protein) incision and Pol β (light blue protein) DNA synthesis activities. (**E**). RECQL4 mediates repair of UV lesions through its interaction with XPA (yellow). XPA works to verify the damage following damage recognition and DNA unwinding by the TFIIH complex (complex of teal proteins). (**F**). RECQL4 interacts with p53 (green protein) to mediate mtDNA synthesis through Polγ (light blue) and to sequester p53 in the mitochondria. However, following DNA damage, the interaction between RECQL4 and p53 is lost and both proteins relocalize to the nucleus to mediate the DNA damage response. Created using biorender.com, accessed on 11 November 2021.

**Table 1 genes-12-01919-t001:** Abbreviations Used.

AP	Apurinic
ATM	Ataxia-telangiectasia mutated
BER	Base excision repair
BGS	Baller-Gerold syndrome
BLM	Bloom syndrome protein
CPD	Cyclobutane pyrimidine dimers
CMG	CDC45-MCM2-7-GINS
DSB	Double-strand break
DSBR	Double-strand break repair
IP	Immunoprecipitation
IR	Ionizing radiation
MLS	Mitochondrial localization sequence
mtDNA	Mitochondria DNA
NER	Nucleotide excision repair
NHEJ	Non-Homologous End Joining
NLS	Nuclear localization sequence
RTS	Rothmund-Thomson syndrome
xRTS	*Xenopus* RECQL4
TIFs	Telomere dysfunction-induced foci
UV	Ultraviolet
WRN	Werner syndrome ATP-dependent helicase

**Table 2 genes-12-01919-t002:** RECQL4 Protein interactions.

Process	Protein	Detection Methods *	Interaction Region	Function	References
Localization	p300	Co-IP, Co-Loc, Pull Down	1–408 aa	RECQL4 cellular localization	[9]
Replication	Cut5	Co-IP	N-terminus	DNA Replication, *Xenopus* Cut5 with xRTS	[48]
	MCM7	Co-IP, MS	ND	DNA Replication	[49]
	MCM10	Co-IP, MS, Pull Down	1–200 aa	DNA Replication, inhibition of RECQL4 helicase activity	[49]
	SLD5	Co-IP, MS	ND	DNA Replication	[49]
Telomere	TRF1	Co-Loc	ND	Telomere maintenance, Stimulates RECQL4 helicase activity	[50]
	TRF2	Co-IP	ND	Telomere maintenance, Stimulates RECQL4 helicase activity	[50]
	WRN	Co-IP	ND	RECQL4 stimulates WRN telomeric D-loop resolution	[50]
DSB Repair	BLM	Co-IP, Y2H	1–471 aa	Increase RECQL4 retention time at DSB sites, BLM stimulation	[51]
	KU70	Co-IP	N-terminus	NHEJ, RECQL4 enhances KU complex DNA binding. KU inhibits RECQL4 helicase activity	[52]
	KU80	Co-IP	ND	NHEJ, RECQL4 enhances KU complex DNA binding. KU inhibits RECQL4 helicase activity	[52]
	MRE11	Co-Loc, IP, Pull Down	N-terminus	HR, DNA end resection	[53]
	RAD51	Co-Loc, Co-IP	ND	DSB Repair	[54]
BER	APE1	Co-Loc	ND	APE1 endonuclease stimulation	[55]
	FEN1	Co-Loc	ND	FEN1 incision stimulation	[55]
	OGG1	Co-IP	N-terminus	Stimulates OGG1 AP lyase activity	[56]
	PARP1	Co-IP, PDS	833–1208 aa	Base excision repair	[57]
	POL β	ND	ND	Stimulates POLβ DNA synthesis activity	[55]
NER	XPA	Co-IP, Co-Loc, Fractionation, Pull Down	ND	Nucleotide excision repair	[58]
Mitochondria	p53	Co-IP, Co-Loc, Fractination	270–400 aa	Sequestering p53 from nucleus, mtDNA synthesis	[44]
	TOM20	Pull Down	13–18 aa	Mitochondrial import	[44]

* Abbreviations: Co-IP, Co-Immunoprecipitation; Co-Loc, Co-Localization; IP, immunoprecipitation; MS, mass spectrometry; ND, not determined; PDS, phage display.

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
