# Peer review of "Role and Regulation of the RECQL4 Family during Genomic Integrity Maintenance"

_genes, 2021, doi:10.3390/genes12121919_

Round 1
Reviewer 1 Report
The Authors present a review on the multifaceted functions of RECQL4 helicase in the maintenance of genome stability. The review briefly introduces the three partially overlapping syndromes caused by biallelic pathogenic variants of the RECQL4 gene, then overviews the model systems used to decipher from yeast to human the RECQL4 functions here outlined in DNA replication, telomere maintenance, DNA repair pathways and mitochondrial integrity.
All sections dedicated to the role of RECQL4 in the DNA replication and repair pathways and mitochondrial maintenance are well planned and provide updated information. Some details on the clinical front, especially on cancer development by RTS individuals with loss-of-function RECQL4 mutations would be useful to attest this most devastating clinical sign driven by disruption of genomic stability and also to discuss the poorly understood effect of RECQL4 overexpression in sporadic cancers.
In detail the following criticisms/suggestions should be taken into account:
Lines 47-48 and Fig 1A- A functional RECQL4 RQC domain has been bioinformatically predicted (Marino F, 2013) and then confirmed to be functional by inductively coupled plasma-atomic emission spectrometry and analysis of site-directed mutants (Mojundar A, 2017).
Figure 2: I would specify Coronal craniosynostosis for BGS , add premature aging for RTS and indicate osteosarcoma and lymphoma shared by RTS and RAPADILINO.
Lines 90- fewer than forty cases with clinical diagnosis of BGS have been reported , but only 11 have been genetically characterized and a few have been followed-up: one of them developed a mid-line lymphoma (Debeljak, 2009; Van Maldergem et al, 2018). There is no ground to support/rule out cancer predisposition in BGS
Lines 96-99 There is a huge number of genes implicated in syndromes presenting craniosynostosis among the clinical signs. The focus of this review is RECQL4 , which biallelic pathogenic variants underlie 3 syndromes with partial overlap. The sentence: “ Perhaps the involvement of FGFR2 and TWIST could explain…..” does convey a misleading message. Saethre-Chotzen syndrome, caused by monoallelic pathogenic variants in TWIST may enter in differential diagnosis with BGS, and the FGFR2 gene when mutated leads to several syndromes a few sharing signs with BGS, but again this is relevant only to formulate the clinical diagnosis of BGS which is not a topic of this review.
Line 113 replace more frequently observed with “most frequently observed” : in the cited Siitonen paper 14 patients were Finnish and only 2 non-Finnish
Line 114 Before discussing the role of RECQL4 in sporadic cancers a few lines on predisposition to cancer of RTS and RAPADILINO individuals with germline RECQL4 pathogenic variants, some associated with cancer outcome (Colombo EA, 2018) should be inserted, mentioning the cancer types, in particular for RTS osteosarcoma at pediatric age and cutaneous epithelial neoplasms later and lymphoma for RAPADILINO. Chromosomal instability, well documented in RTS patients (Miozzo, 1998; Beghini, 2003), is a powerful driver of unbalanced cells undergoing neoplastic transformation in the tissues most sensitive to RECQL4 loss of function.
For better readability Table 1 and Table 2 should start at page 4 and 5, respectively and not be divided each in two consecutive pages
There is something wrong in Headings 3 and 4 : both read “Conserved role for RECQL4 family during DNA replication”: should heading 4 be “Conserved role for the RECAL4 family during telomeric DNA replication??
Please reset Figure 2, which should be Fig.3 and its legend : the 3 top panels don’t have the letters (A, B, C) and the bottom panels should have D, E, and F letters instead of A, B, C . The colors often don’t correspond to; the legend has some gaps in some words. Quadraplexes should be quadruplexes.
Heading 7: Role of RECQL4 in mitochondrial maintenance may be placed before Conclusions, in order to keep together the different DNA repair pathways
Line 464 : add “role” after a direct
Line 487 HRQ1 should be Hrq1
Conclusion : considering the potential role of REQL4 in crosslink repair, till better understanding how DNA crosslinks are repaired, recommendations should be given to oncologists on chemotherapeutics of RTS patients developing cancer.
Author Response
Response to Reviewer 1 Comments:
We thank the reviewer for their careful reading of the manuscript and insightful comments. We have now adjusted the texts and figures according to their suggestions.
Lines 47-48 and Fig 1A- A functional RECQL4 RQC domain has been bioinformatically predicted (Marino F, 2013) and then confirmed to be functional by inductively coupled plasma-atomic emission spectrometry and analysis of site-directed mutants (Mojundar A, 2017).
We thank the reviewer for this important oversight and have now added the RQC domain to Figure 1A and 1B and adjusted the text accordingly.
Figure 2: I would specify Coronal craniosynostosis for BGS , add premature aging for RTS and indicate osteosarcoma and lymphoma shared by RTS and RAPADILINO.
We have now added these symptoms to Figure 2 and also in the text.
Lines 90- fewer than forty cases with clinical diagnosis of BGS have been reported , but only 11 have been genetically characterized and a few have been followed-up: one of them developed a mid-line lymphoma (Debeljak, 2009; Van Maldergem et al, 2018). There is no ground to support/rule out cancer predisposition in BGS.
We have adjusted the text accordingly.
Lines 96-99 There is a huge number of genes implicated in syndromes presenting craniosynostosis among the clinical signs. The focus of this review is RECQL4 , which biallelic pathogenic variants underlie 3 syndromes with partial overlap. The sentence: “ Perhaps the involvement of FGFR2 and TWIST could explain…..” does convey a misleading message. Saethre-Chotzen syndrome, caused by monoallelic pathogenic variants in TWIST may enter in differential diagnosis with BGS, and the FGFR2 gene when mutated leads to several syndromes a few sharing signs with BGS, but again this is relevant only to formulate the clinical diagnosis of BGS which is not a topic of this review.
We agree with the reviewer that this could convey a misleading message, we have now removed the sentence.
Line 113 replace more frequently observed with “most frequently observed” : in the cited Siitonen paper 14 patients were Finnish and only 2 non-Finnish
Corrected.
Line 114 Before discussing the role of RECQL4 in sporadic cancers a few lines on predisposition to cancer of RTS and RAPADILINO individuals with germline RECQL4 pathogenic variants, some associated with cancer outcome (Colombo EA, 2018) should be inserted, mentioning the cancer types, in particular for RTS osteosarcoma at pediatric age and cutaneous epithelial neoplasms later and lymphoma for RAPADILINO. Chromosomal instability, well documented in RTS patients (Miozzo, 1998; Beghini, 2003), is a powerful driver of unbalanced cells undergoing neoplastic transformation in the tissues most sensitive to RECQL4 loss of function.
We thank the reviewer for this insightful comment, we have added a few lines discussing how RECQL4 loss of function could drive cancer.
For better readability Table 1 and Table 2 should start at page 4 and 5, respectively and not be divided each in two consecutive pages
Fixed.
There is something wrong in Headings 3 and 4 : both read “Conserved role for RECQL4 family during DNA replication”: should heading 4 be “Conserved role for the RECAL4 family during telomeric DNA replication??
We apologized for the error which has now been corrected.
Please reset Figure 2, which should be Fig.3 and its legend : the 3 top panels don’t have the letters (A, B, C) and the bottom panels should have D, E, and F letters instead of A, B, C . The colors often don’t correspond to; the legend has some gaps in some words. Quadraplexes should be quadruplexes.
Corrected.
Heading 7: Role of RECQL4 in mitochondrial maintenance may be placed before Conclusions, in order to keep together the different DNA repair pathways
We agree with the reviewer and have moved the mitochondrial section to keep the DNA repair pathways together.
Line 464 : add “role” after a direct
Corrected.
Line 487 HRQ1 should be Hrq1
Since we are referring to a wild-type yeast gene, the proper nomenclature is to keep the gene name capitalized and italicized.
Conclusion : considering the potential role of REQL4 in crosslink repair, till better understanding how DNA crosslinks are repaired, recommendations should be given to oncologists on chemotherapeutics of RTS patients developing cancer.
We have now modified the sentence to be clearer, “Since DNA crosslinking agents are used as chemotherapeutics, understanding how RECQL4 repairs crosslinks could be informative for clinicians.”
Reviewer 2 Report
This is a comprehensive review on REQL4 helicases.
Minor comments:
Lane 27: Indicate: Bloom syndrome protein (BLM) / Werner syndrome ATP-dependent helicase (WRN)
Lane 44: Change to: One critical member of this helicase family is is encoded by the RECQL4 gene, which is located on chromosome 44 8q24.3. The resulting 1208 amino acid sized protein has a conserved core domain, however, ………..
Lane 56: change to: (discussed below; Figure 1B).
Lanes 124 to 131: Please clarify better the different impact of RECQL4 on cancer cell resistance to chemotherapeutic agents.
Throughout the text: Clarify the use of protein/gene name abbreviations.
Lane 159: Please clarify if yeast genes yeast genes can have similar hydrophobic regions
Lanes 284-286: Please clarify the apparently contradictory result that the lack or excess of Hrq1 leads to telomere extension.
Lane 295: better: …a single, unrepaired DSB…..
Lane 439: skip: via invasion assays.
Lane 443: better: …key role in the repair of oxidative damage in mitochondrial DNA.
Lane 502: change: …..crosslinks are repaired…..
Figure 1: Protein length should be indicated by using amino acids (aa).
Figure 2: Please clarify the bugs in the Figure legend.
Reference 29: Clarify the formatting
I recommend the authors to carefully check the review for minor bugs.
Author Response
Response to Reviewer 2 Comments:
We thank the reviewer for their careful reading of the manuscript, and we have now edited the manuscript according to their suggestions as detailed below.
Minor comments:
Lane 27: Indicate: Bloom syndrome protein (BLM) / Werner syndrome ATP-dependent helicase (WRN)
Corrected.
Lane 44: Change to: One critical member of this helicase family is is encoded by the RECQL4gene, which is located on chromosome 44 8q24.3. The resulting 1208 amino acid sized protein has a conserved core domain, however, ………..
Corrected.
Lane 56: change to: (discussed below; Figure 1B).
Corrected.
Lanes 124 to 131: Please clarify better the different impact of RECQL4 on cancer cell resistance to chemotherapeutic agents.
We have clarified the impact that RECQL4 has in driving resistance to cisplatin.
Throughout the text: Clarify the use of protein/gene name abbreviations.
Corrected.
Lane 159: Please clarify if yeast genes yeast genes can have similar hydrophobic regions
We have clarified that Hrq1 has similar hydrophobic regions to human RECQL4.
Lanes 284-286: Please clarify the apparently contradictory result that the lack or excess of Hrq1 leads to telomere extension.
We thank the reviewer for this comment, we have clarified it in the text.
Lane 295: better: …a single, unrepaired DSB…..
Corrected.
Lane 439: skip: via invasion assays.
Corrected.
Lane 443: better: …key role in the repair of oxidative damage in mitochondrial DNA.
Corrected.
Lane 502: change: …..crosslinks are repaired…..
Corrected.
Figure 1: Protein length should be indicated by using amino acids (aa).
Corrected.
Figure 2: Please clarify the bugs in the Figure legend.
Corrected.
Reference 29: Clarify the formatting
Corrected.
I recommend the authors to carefully check the review for minor bugs.
We have now fixed the minor bugs throughout the manuscript.